# LATENT SPACE UNIFORMIZATION IN GENERATION

## ABSTRACT

Latent space transformation is a core topic in generative model research and is crucial for understanding and controlling the generative process. This study proposes the concept of latent space uniformization inspired by Coulomb's law, named ULatent. This concept provides a canonical representation of the latent space and facilitates sampling and aligning elements in both single-domain and cross-domain generative scenarios. Specifically, we model data points in a two-dimensional latent space as charged particles driven by Coulomb forces (electrostatic dynamics). The repulsive forces between them form a uniform distribution in the latent space, simulating the phenomenon where equally charged particles reach equilibrium. The uniformization of the original data enhances latent space structure, particularly by eliminating gaps between isolated clusters. For semantically overlapping clusters, pre-translation operations are required. By integrating geometric mapping techniques, we achieve precise alignment of uniformly distributed data across both single-modal and multi-modal domains, thereby simultaneously improving sampling efficiency and generation accuracy. All these conclusions are validated through multi-dataset experiments and ablation studies.

## 1 INTRODUCTION

The power of deep learning lies in learning a meaningful latent space that captures semantic concepts. Once this space is established, transforming the distributions within it becomes the primary lever for achieving domain alignment, data fusion, knowledge transfer, and controlled generation. Deep generative models target connecting the efficiency of simple distributions (ease of sampling) with the expressiveness of complex distributions (generating rich data), such as normalizing flow (Weng, 2018), VAE (Kingma & Welling, 2013), GAN (Goodfellow et al., 2014), and diffusion model (Ho et al., 2020). Standard normal distribution (Gaussian distribution), uniform distribution, and mixture distribution are usually used as priors.

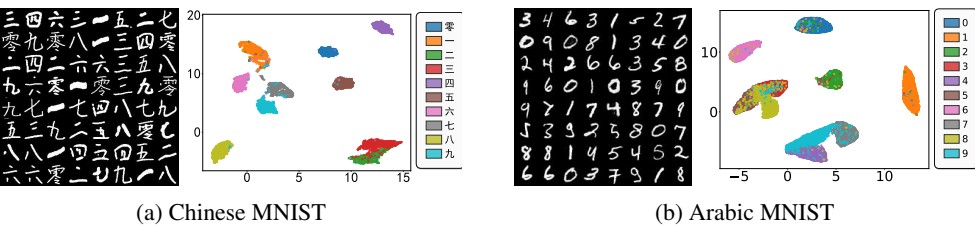

(a) Chinese MNIST            (b) Arabic MNIST

Figure 1: 2D latent space via uniform manifold approximation and projection (UMAP).

Here, we focus on transforming data distribution to be uniform and create a canonical representation of the latent space formed by the raw data samples. Its core essence is to eliminate inherent data imbalances, thereby enhancing model performance, stability, and interpretability, and assisting sampling procedure in generation. In a 2D context, uniform distribution exhibits both simple statistical properties and clearly distinguishable geometric structures. From a geometric perspective, the point set formed in the 2D latent space—derived from original data samples through encoding and dimensionality reduction—exhibits irregular boundary shapes and non-uniform, clustered density distributions (see Fig. 1). This complex morphology compels us to seek more intuitive ways to visualize the latent space's structure. The idea of uniformization within regular boundaries naturally emerges, thereby establishing a canonical representation in both shape and distribution (see Fig. 2).

Figure 2: Latent space transformations and canonical representation.

This transformation mechanism enables efficient geometric processing within latent spaces and across different latent spaces. It facilitates controlled sampling of latent spaces and achieves cross-domain precise latent space alignment in both single-modal and multi-modal scenarios.

**Contributions.** This work firstly introduces the concept of *latent space uniformization* (ULatent), emphasizing the preservation of original samples and their efficient visualization. In detail,

1. Based on Coulomb's law, the latent space undergoes a uniformization process, treating the samples as electrically charged particles that repel each other until a stable state is achieved.

2. Efficient sampling and spherical interpolation strategies are implemented on uniformly distributed original samples, achieving controllable generation with high precision and quality.

## 2  RELATED WORK

This work focuses on cross-domain generative artificial intelligence, an area centered on three key technologies: 1) latent space representation, 2) domain transformation in latent space, and 3) latent space alignment. We now briefly review related works through the lens of these technologies.

### 2.1  LATENT GENERATIVE MODEL

Latent generative models learn the underlying data distribution to enable sample generation via a generator network $G$ that maps low-dimensional latent vectors $z$ from a prior distribution (e.g., Gaussian) to the data space $\mathcal{X}$: $G : \mathcal{Z} \to \mathcal{X}$. The goal is to obtain a smooth and interpretable latent space $\mathcal{Z}$ where every point decodes to a meaningful sample. The following are some commonly used basic generative models and improved generative models: An autoencoder (AE) (Hinton & Zemel, 1993) uses an encoder $g_w$ to map input $x$ to a latent representation $z = g_w(x)$, and a decoder $f_\theta$ to reconstruct $x \approx f_\theta(z)$. However, the AE latent space is often irregular, containing low-density regions that decode to meaningless outputs. The variational autoencoder (VAE) (Kingma & Welling, 2013) introduces probabilistic latent constraints to improve smoothness and support generation, though often at the cost of blurry outputs due to regularization. Generative adversarial networks (GANs) (Goodfellow et al., 2014) train a generator $G$ and discriminator $D$ adversarially, where $G$ maps noise $z$ to data space and $D$ distinguishes real from generated samples. GANs yield high-fidelity samples but suffer from training instability and mode collapse. Diffusion models (Ho et al., 2020) generate data via iterative noise addition and denoising, operating over high-dimensional intermediate latent states that lack semantic interpretability. AE-OT (An et al., 2019) uses optimal transport to structure the latent space, closely matching a prior distribution to reduce mode collapse and improve coverage. AE-OT-GAN (An et al., 2020) extends this in two stages: first learning a data-informed latent distribution $p(z)$ via AE-OT, then training a GAN from samples in $p(z)$. This hybrid improves stability and sample quality while mitigating mode collapse.

### 2.2  LATENT SPACE TRANSFORMATION

Once a well-structured latent space is learned, latent space transformation techniques can be applied to exert fine-grained semantic control over the generation process. This approach is grounded in the key hypothesis that geometric directions in the latent space correspond to meaningful semantic attribute changes in the data. Given a pre-trained generative model $G$ and its latent space $z$, semantic

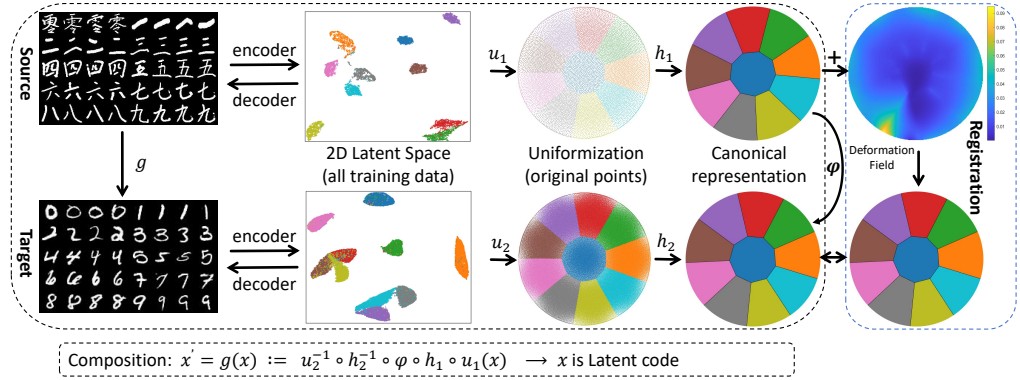

Composition: $x' = g(x) := u_2^{-1} \circ h_2^{-1} \circ \varphi \circ h_1 \circ u_1(x) \longrightarrow x$ is Latent code

Figure 3: Cross-domain generative model. $u_1, u_2$ – uniformization; $h_2, h_2$ – graph-constrained harmonic map (straightening); $\varphi$ – graph-constrained harmonic registration (alignment).

latent space editing $\hat{z}$ is achieved through vector arithmetic:

$$\hat{z} = z + \alpha \overrightarrow{n}, \tag{1}$$

where $\overrightarrow{n}$ is a direction vector representing a target structure, and $\alpha$ is a scalar controlling the editing strength. Several recent studies have explored latent space deformation for enhanced controllability. GeoLatent (Yang et al., 2023a) incorporates geometric inductive biases into the latent space for deformable shape generation, constructing a manifold endowed with geometric properties—such as geodesic distance—to support disentangled and controllable shape editing. GLASS (Muralikrishnan et al., 2022) improves the generalization of shape spaces through geometric-aware latent augmentation, generating diverse and realistic training samples to bolster the robustness of downstream models. GMapLatent (Zeng et al., 2025) transforms the latent space into a canonical parameter domain via barycentric translation, optimal transport-based merging, and constrained harmonic mapping, followed by geometric registration with cluster constraints in the canonical domain. These transformation techniques, all fundamentally leveraging the geometric relationships among original samples, facilitate disentangled and controllable editing of generated content, playing a pivotal role in bridging static generative models with interactive AI applications.

## 2.3 LATENT SPACE ALIGNMENT

Latent space alignment enables cross-domain understanding and generation by establishing semantically consistent mappings between independently learned latent spaces. The goal is to learn a mapping function $\varphi_{A \to B}$ that projects points from $\mathcal{Z}_A$ to corresponding points in $\mathcal{Z}_B$, preserving semantic meaning across domains. Domain Adaptation (DA) methods often align feature or latent space distributions. Some approaches introduce structural constraints for multi-class alignment: for example, Gu et al. (2022) uses keypoint constraints within an optimal transport (OT) framework, and Yang et al. (2023b) applies prototypical constraints under OT for universal domain adaptation. While these methods achieve coarse alignment, they often lack precise structural correspondence at the cluster level. GMapLatent (Zeng et al., 2025) addresses this gap by reformulating cluster alignment as a surface registration problem. The method first employs semi-discrete optimal transport to disperse latent clusters into a rectangular parameter domain, thereby unifying their metric structure. Subsequent geometric mapping aligns the resulting surfaces, mitigating mode collapse and resolving latent space discontinuities commonly found in autoencoder-derived representations. However, due to the limited representational capacity of the parametric domain, the mapping between points in the canonical representation and the original latent clusters is not bijective. This many-to-one correspondence leads to local information loss during alignment. Therefore, in Section 3, we propose a *latent space uniformization* (ULatent) method to address the issue of latent space alignment.

## 3 METHOD

ULatent enables cross-domain image generation by aligning latent spaces into a shared 2D semantic structure (see Fig. 3). It maps different domains onto a standardized manifold, maintaining se-

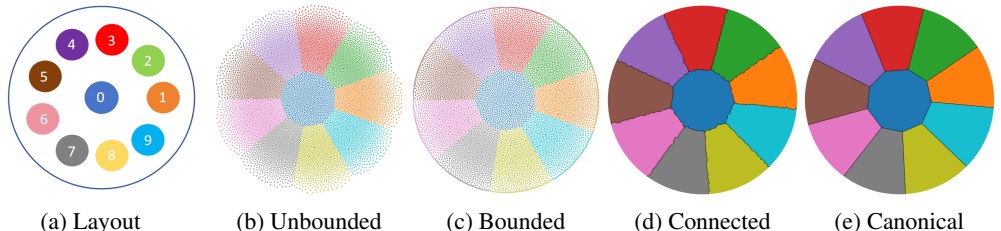

Figure 4: Single-domain generative model.

mantic distinction while enabling smooth geometric transitions. Through barycentric translation, uniformization, and triangulation, it forms a canonical latent space suitable for high-quality interpolation. Graph-constrained harmonic registration ensures structurally consistent and visually coherent generation. Next, we will elaborate on three key core points: 1) the canonical representation of the latent space in a single domain based on latent space uniformization, 2) the alignment of canonical spaces across domains, and 3) the generation of cross-domain samples with interpolation strategy.

### 3.1 CANONICAL REPRESENTATION OF SINGLE DOMAIN

This process aims to project the high-dimensional latent codes extracted by the encoder into a low-dimensional space. This target space is a canonical space, featuring a clear geometric structure and favorable mathematical properties (see Fig. 4).The constructing pipelines are as follows: 1) 2D visualization. We map the latent codes to a 2D space.2) Barycentric translation. We perform preliminary separation of different semantic categories in the latent space, laying the geometric foundation for subsequent operations (see Fig. 5a), the details are presented in Section 3.1.1. 3) Uniformization. Subsequently, we simulate the repulsion process using Coulomb forces by assigning equal charges to all data points with convex boundary constraint. Upon reaching a stable state, they exhibit a uniform distribution (see Figs. 5b-5c). The details are presented in Section 3.1.2. 4) Straightening. We finally apply the graph-constrained harmonic map (Yang et al., 2018) onto the latent space domain (represented in triangular mesh), to straighten the curvy graph formed by cluster boundaries into convex subdivision (see Fig. 5d-5e) while preserving mapping smoothness as much as possible.

| (a) Layout | (b) Unbounded | (c) Bounded | (d) Connected | (e) Canonical |

Figure 5: Multi-modal uniformization.

### 3.1.1 BARYCENTRIC TRANSLATION

For the 2D data following dimensionality reduction via UMAP, assume there are $n$ modality categories. When $n > 3$ (see Fig. 5a), one cluster center is placed at position $(0,0)$, while the remaining $n-1$ cluster centers are distributed on a disk of radius $R$ according to the ratio of the number of samples in each category. When $n \leq 3$ (see Fig. 6 for a single cluster), the cluster centers are distributed on a disk by their size ratio. For clusters of equal size, the centroid coordinates are given by

$$\mathbf{c}_i = \begin{cases} \left(R \cdot \cos\left(\frac{2\pi i}{n}\right), R \cdot \sin\left(\frac{2\pi i}{n}\right)\right) & \text{if } n \leq 3 \\ (0,0) & \text{if } n > 3 \text{ and } i = 0 \\ \left(R \cdot \cos\left(\frac{2\pi(i-1)}{n-1}\right), R \cdot \sin\left(\frac{2\pi(i-1)}{n-1}\right)\right) & \text{if } n > 3 \text{ and } i \geq 1 \end{cases} \quad (2)$$

where $n$ denotes the number of categories; $R$ is a tunable parameter representing the radius of the disk; $i = 0, 1, ..., n-1$ is the category index; $c_i$ is the centroid coordinate of the $i$-th category.

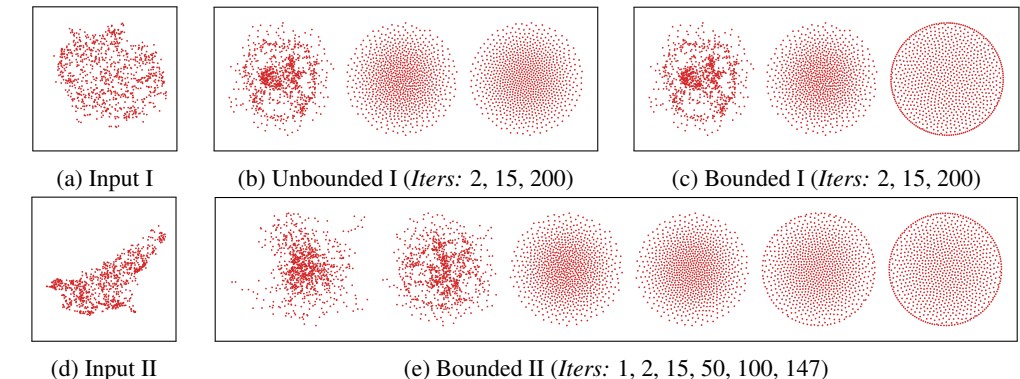

(a) Input I     (b) Unbounded I (*Iters:* 2, 15, 200)     (c) Bounded I (*Iters:* 2, 15, 200)

(d) Input II     (e) Bounded II (*Iters:* 1, 2, 15, 50, 100, 147)

Figure 6: Single-modal uniformization. Results at different iterations (*Iters*) are given.

### 3.1.2 UNIFORMIZATION

The uniformization is inspired by Coulomb's law (see Theorem 3.1).

**Theorem 3.1** (Coulomb's Law). *Coulomb's law quantitatively describes the magnitude and direction of the force of interaction between two stationary point charges,*

$$F = k_e \frac{q_1 q_2}{r^2}, \tag{3}$$

*where $F$ denotes the magnitude of the electrostatic force between charges $q_1$ and $q_2$, $r$ denotes the distance between the two charges, and $k_e$ is Coulomb's constant.*

It causes data points to mimic charged particles of equal charge reaching equilibrium. The computation is illustrated in Algorithm 1 (see Appendix A.1). Figure 6 demonstrates the optimization process of single-modal clusters, one relatively regular cluster and one irregularly shaped (see Figs. 6a, 6d). The unbounded and bounded (with exterior circular boundary constraint) cases remain the same before the particles reach boundary (see Figs. 6b- 6c). The algorithm demonstrates stability and efficiency even when processing irregularly shaped clusters (see Fig. 6e).

For multi-modal case (see Fig. 5), barycentric translations are performed before uniformization. The particle repulsion model drives latent points to disperse within their respective semantic clusters. The system converges readily to a stable state of minimal energy, ultimately forming a canonical latent space characterized by uniform distribution and distinct boundaries between them.

The convergence of the algorithm is guaranteed by Theorem 3.2 (see Appendix A.2 for proof):

**Theorem 3.2** (Convergence). *We first define the potential energy*

$$U(P) = \alpha \sum_i \sum_{j \in \mathcal{N}_k(i)} \frac{1}{2\sqrt{\beta}} \arctan\left( \frac{\|p_i - p_j\|}{\sqrt{\beta}} \right), \tag{4}$$

*where $\mathcal{N}_k(i)$ denotes the $k$-nearest neighbors of point $i$. A lower $U(P)$ indicates a more uniform point layout within each cluster.*

*Consider the iteration*

$$P^{t+1} = \Pi_K \left( P^t - \eta \nabla U(P^t) \right), \tag{5}$$

*where $\Pi_K$ is the projection onto the feasible set $K$ (e.g., the circular boundary). If $\nabla U(P)$ is $L$-Lipschitz continuous on $K$ and the step size satisfies $0 < \eta \leq 1/L$ (small enough to avoid oscillations), then the sequence $\{U(P^t)\}$ decreases monotonically and converges to a finite equilibrium value $U_\infty$.*

The repulsive force has finite range at short distances and decays as $O(1/r^2)$ for large $r$, preventing point overlap and ensuring numerical stability. The iteration is equivalent to gradient descent on the potential energy and converges to equilibrium under suitable step sizes, with boundary projection not affecting convergence. Each iteration costs $O(N \log N + kN)$ for KD-tree search and force computation, giving total complexity $O(T_{\max} \cdot (N \log N + kN))$.

The unified algorithm naturally adapts to datasets with imbalanced category sizes by adjusting only the angular width of each sector; thus, even when the scale of categories changes, the final unified configuration maintains a stable topology (see Fig. 7).

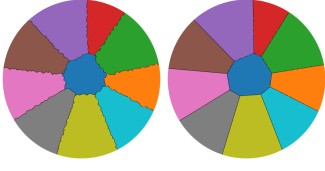

(a) Original    (b) Canonical

Figure 7: Imbalanced volumes.

## 3.2 LATENT SPACE ALIGNMENT

The source and target domains correspond to distinct data distributions; the goal of latent space alignment is to register their latent representations under given structural constraints. With the specified category pairing constraints (cluster constraints), we perform the barycentric translations consistently if there are multiple categories. For $n > 3$ cases, one cluster is set to the center and others surround it, which generates degree-three junctions in cluster graph (see Fig. 5). This guarantees the same topology in both source and target latent space transformations. After we obtain the canonical convex-subdivision domains for both source and target, we compute a graph-constrained harmonic map betwee them to generate the precise geometric registration. The computational algorithm can be found in (Yang et al., 2018).

## 3.3 CROSS-DOMAIN GENERATION

In the generative task for the test set, the input data are first mapped into high-dimensional latent representations via a pre-trained AE encoder, followed by dimensionality reduction to two dimensions using UMAP. Leveraging the triangulation constructed during training, we identify the semantic triangle containing the projected point and compute its barycentric coordinates, which are then mapped to the canonical space of the target domain. Through inverse mapping, the corresponding high-dimensional latent representations in the target domain are recovered, yielding three points associated with the source semantics. In the high-dimensional latent space, interpolation is performed on these three points. We have designed two interpolation methods: 1) linear interpolation, as detailed in Section 3.3.1; 2) spherical linear interpolation (Slerp), as detailed in Section 3.3.2. This enables continuous sampling on the semantic manifold, thereby generating new samples with clear semantic consistency.

### 3.3.1 LINEAR INTERPOLATION

We first employ a linear interpolation method to reconstruct the latent space (see Fig. 8). The basic idea is to use the latent encodings of the vertices of a semantic triangle and their barycentric coordinates. Specifically, in the canonical space, for any semantic triangle defined by triangulation, its three vertices correspond to the latent encoding vectors $z_A, z_B$ and $z_C$ in the high-dimensional latent space. Let the barycentric coordinates of the target point within this triangle be $(\alpha, \beta, \gamma)$, satisfying $\alpha + \beta + \gamma = 1$ and $\alpha, \beta, \gamma \geq 0$. Then the interpolated representation $z$ of this point in the high-dimensional latent space can be obtained through the linear combination: $z = \alpha z_A + \beta z_B + \gamma z_C$.

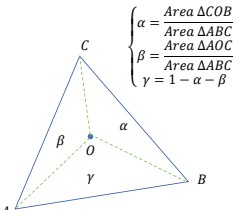

Figure 8: Barycentric coordinates.

### 3.3.2 SPHERICAL LINEAR INTERPOLATION

Since high-dimensional latent manifolds are usually non-Euclidean, the latent encodings generated by linear interpolation often deviate from the true data manifold, leading to blurring and structural distortion in the decoded images, thereby degrading semantic consistency and visual realism. To address this issue, we introduce the spherical linear interpolation method to better preserve the geometric properties and distribution constraints of latent vectors. This approach treats latent vectors as points on a high-dimensional sphere, employing interpolation along spherical geodesic paths to ensure generated points remain near the manifold. For three-vertex interpolation scenarios, we adopt a hierarchical Slerp strategy. Since Slerp is performed along the unit vector, it alters the direction but not the magnitude. Therefore, when performing Slerp, we apply linear interpolation to the magnitude. For two unit vectors $\mathbf{a}$ and $\mathbf{b}$, the following formula holds:

$$\begin{cases} \text{Slerp}\,(\mathbf{a}, \mathbf{b}, t) = [\sin((1-t)\theta)/\sin\theta] \cdot \mathbf{a} + [\sin(t\theta)/\sin\theta] \cdot \mathbf{b} \\ \theta = \arccos(\mathbf{a} \cdot \mathbf{b}) \end{cases} \tag{6}$$

For three unit vectors $\mathbf{d}_0$, $\mathbf{d}_1$, $\mathbf{d}_2$ and weights $\alpha$, $\beta$, $\gamma$, the interpolation formula is:

$$\mathbf{d} = \mathrm{Slerp}\left(\mathrm{Slerp}\left(\mathbf{d}_0, \mathbf{d}_1, \alpha/(\alpha + \beta)\right), \mathbf{d}_2, \alpha + \beta\right). \tag{7}$$

Therefore, for the three latent codes $\mathbf{z}_0$, $\mathbf{z}_1$, and $\mathbf{z}_2$, the interpolation formula is:

$$\mathbf{z} = \mathbf{d} \cdot \mathbf{r} = \mathrm{Slerp}\left(\mathrm{Slerp}\left(\frac{\mathbf{z}_0}{\|\mathbf{z}_0\|}, \frac{\mathbf{z}_1}{\|\mathbf{z}_1\|}, \frac{\alpha}{\alpha + \beta}\right), \frac{\mathbf{z}_2}{\|\mathbf{z}_2\|}, \alpha + \beta\right) \cdot \left(\alpha\|\mathbf{z}_0\| + \beta\|\mathbf{z}_1\| + \gamma\|\mathbf{z}_2\|\right). \tag{8}$$

## 4 EXPERIMENT AND RESULTS

### 4.1 EXPERIMENT SETTINGS

Our experiments focus on cross-domain generation, where domain adaptation is achieved by precisely aligning the latent spaces of the source and target domains. Transfer learning is subsequently applied to map the source domain to the target domain, enabling the generation of target-domain samples. We evaluate the proposed ULatent model on two types of datasets—binary handwritten digits and natural color images—and compare its performance against existing methods. Evaluation is conducted along two key dimensions: image quality, assessed using the Fréchet Inception Distance (FID) to measure the distributional divergence between generated and real images (lower FID indicates higher quality), and semantic accuracy, quantified by the match rate between generated images and their ground-truth categories in the target domain, as determined by a pre-trained LeNet classifier.

Table 1: Comparison with state-of-the-arts methods: $Acc$ - Accuracy (%).

| Method | Digit | | Animal | |
|---|---|---|---|---|
| | FID ↓ | Acc ↑ | FID ↓ | Acc ↑ |
| Cycle-GAN (Zhu et al., 2017) | 6.99 | 22.72 | 78.56 | 30.27 |
| TCR (Mustafa & Mantiuk, 2020) | 6.90 | 36.21 | 342.48 | 33.33 |
| W2GAN (Korotin et al., 2019) | 12.04 | 34.21 | 121.86 | 28.40 |
| OT-ICNN (Makkuva et al., 2020) | 14.37 | 29.12 | 126.43 | 34.67 |
| KPG-RL (Gu et al., 2022) | **6.54** | 76.14 | 81.02 | 77.27 |
| GMapLatent (Zeng et al., 2025) | 13.75 | 94.50 | **76.79** | 86.15 |
| **ULatent (ours)** | 13.72 | **95.40** | 81.09 | **87.85** |

### 4.2 COMPARISON WITH STATE-OF-THE-ARTS METHODS

As shown in Table 1, our ULatent model achieves superior semantic consistency while generating FID comparable to mainstream generative models, with classification accuracy significantly surpassing existing approaches. Specifically, methods such as Cycle-GAN (Zhu et al., 2017), TCR (Mustafa & Mantiuk, 2020), W2GAN (Korotin et al., 2019), and OT-ICNN (Makkuva et al., 2020) struggle to achieve accurate category-level transformations due to their failure to explicitly model cross-domain semantic correspondences, resulting in limited recognition accuracy. The KPG-RL (Gu et al., 2022) approach improves semantic alignment to some extent through keypoint guidance, yet its performance remains highly dependent on the accuracy of keypoint localisation. Misalignments near semantic cluster boundaries or across domains can lead to erroneous matches, thereby constraining overall precision. The GMapLatent (Zeng et al., 2025) method achieves uniform registration via optimal transport, albeit with high computational complexity. In contrast, ULatent constructs a normed space that preserves structural integrity, enabling precise, continuous, and one-to-one registration of cross-domain semantic clusters. This fundamentally enhances the controllability and semantic consistency of cross-domain transformations. Concurrently, its spherical interpolation strategy effectively constrains generated samples to reside near the target manifold, further improving the authenticity and discriminative power of the generated results. The combined enhancement

of semantic alignment capability and manifold-aware interpolation mechanisms collectively drives the overall performance improvement of the model. As shown in Fig. 9, our ULatent method outperforms Cycle-GAN in accuracy and yields clearer images via spherical interpolation, which remains within the data manifold. Unlike Cycle-GAN and OT, our approach avoids mode collapse and translation errors. Quantitative results in Table 1 confirm that ULatent achieves an FID of 13.72 and an accuracy of 95.40%, demonstrating superior performance in image translation. We further validate continuous curve-to-curve generation, which requires coherent mapping of entire latent curves rather than discrete points. Without uniformization, curves may cross category boundaries, causing semantic breaks or mode collapse. Our method optimizes latent boundaries via uniformization, enabling smooth cross-domain curve mapping and consistent image-sequence generation—an advantage not seen in existing methods (Fig. 10).

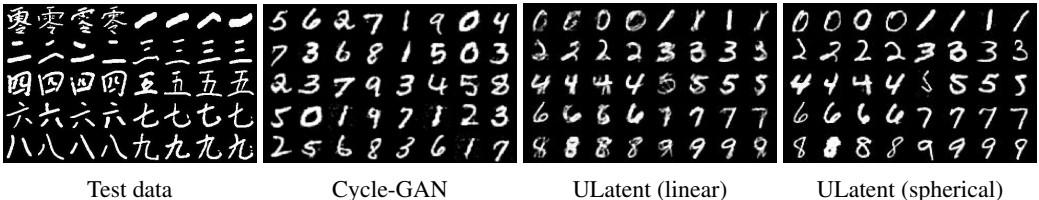

| Test data | Cycle-GAN | ULatent (linear) | ULatent (spherical) |

Figure 9: Comparison of MNIST image translation results.

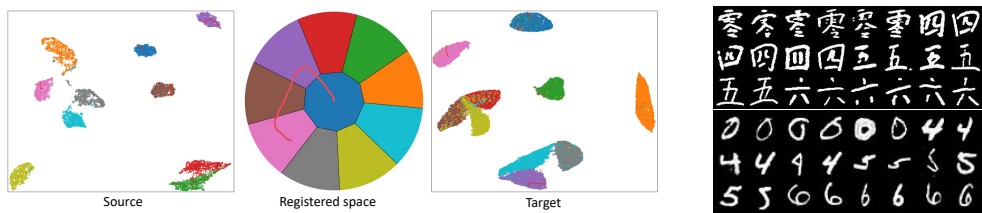

Source    Registered space    Target

Figure 10: Curve-to-curve translation through ULatent.

Generalization to complex data is verified using the AFHQ dataset, with lions, tigers, and wolves as source domains and cats, foxes, and leopards as targets. The decoder mirrors a BigGAN-style generator, and the encoder its symmetric counterpart. ULatent maintains strong performance in this challenging setting, achieving a recognition accuracy exceeding 87.85% and an FID of 81.09, indicating high visual quality and distribution alignment (see Fig. 11).

### 4.3 ABLATION STUDIES

We conducted systematic ablation experiments on the two core modules within the ULatent method—canonical space representation and graph-constrained harmonic registration—with results presented in Table 2. Firstly, we employed the baseline method DirectAlign, which directly aligns latent encodings between source and target domains. Experimental findings indicate that this approach yields higher FID values than ULatent alongside lower classification accuracy, demonstrating that simple direct alignment struggles to achieve high-quality cross-domain semantic mapping. Secondly, we incorporated a canonical space representation module into DirectAlign. Results demonstrate that canonical space representation significantly enhances generation quality and semantic consistency, outperforming direct alignment in both accuracy and FID metrics, though still falling short of the full ULatent model. Finally, we introduced the graph-constrained harmonic registration module, constituting the complete ULatent model. Across both datasets, this model achieved classification accuracy improvements of 10.45% and 6.60%, respectively compared to the uniformization-only approach, while further reducing FID. These results consistently demonstrate that both canonical space representation and graph-constrained harmonic registration modules make significant contributions to enhancing cross-domain generation performance. Their combination maximizes cross-domain semantic alignment and optimizes generation quality.

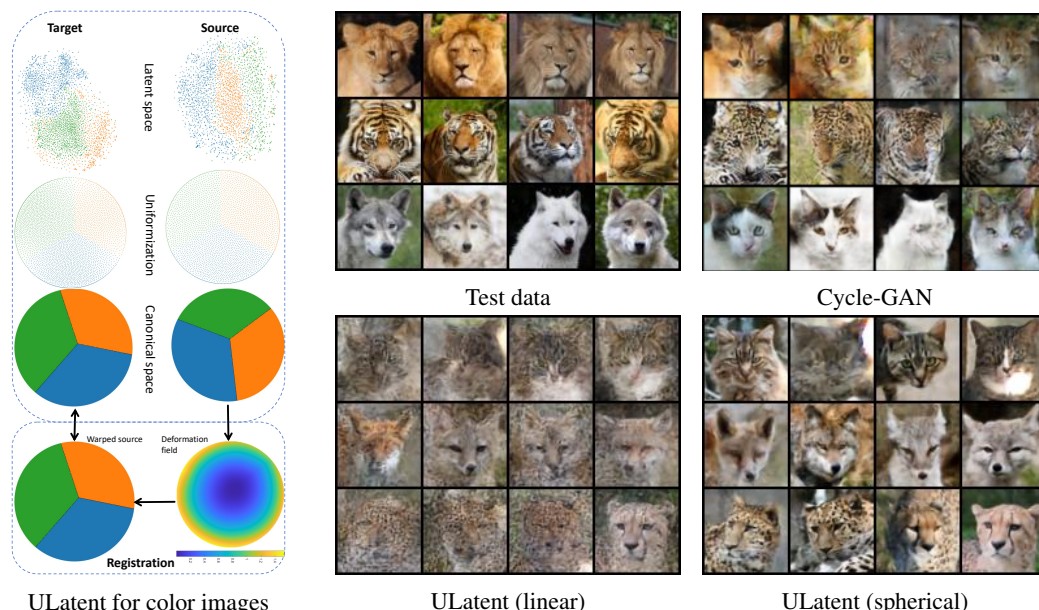

Figure 11: ULatent translation for color images.

Table 2: Ablation. CR-Canonical representation; GCHR-graph-constrained harmonic registration.

| Method | Digit FID ↓ | Digit Acc ↑ | Animal FID ↓ | Animal Acc ↑ |
|---|---|---|---|---|
| DirectAlign (uniformization, *w/o CR; w/o GCHR*) | 15.51 | 84.95 | 76.48 | 81.25 |
| DirectAlign (uniformization, *w CR; w/o GCHR*) | 14.15 | 89.17 | **74.97** | 82.37 |
| **ULatent** (uniformization, *w CR & GCHR*) (**ours**) | **13.72** | **95.40** | 81.09 | **87.85** |

## 4.4 DISCUSSION

The ULatent method differs from traditional approaches that perform augmentation directly in the original image space. By implementing interpolation and transformation on a canonical latent manifold, it ensures generated samples not only exhibit visual realism but also maintain semantic controllability. Experimental results demonstrate that this strategy significantly enhances the generalization capability and cross-domain consistency of generative models, as validated by the simultaneous optimization of accuracy and FID metrics. Notably, ULatent reveals a potential research pathway: treating data augmentation as a structured, semantic manifold operation rather than a simple image transformation. This approach is particularly applicable in domains requiring strict semantic consistency, such as medical imaging and remote sensing imagery.

## 5 CONCLUSION

This work proposes a novel framework name ULatent, which realizes latent space uniformization—inspired by Coulomb's law by modeling latent points as charged particles—for cross-domain generation. By integrating barycentric translation (especially essential in multi-modal scenarios) with geometric mapping operations, we establish a canonical representation of the complete latent space (composed of original data points) and achieve precise alignment between latent spaces. This approach enhances controllability by eliminating gaps between isolated clusters, effectively mitigating mode collapse while preserving semantic consistency, thereby laying a robust foundation for generative modeling. This research pioneers a new paradigm for semantically controllable generation based on geometric latent space regularization.

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

# A APPENDIX

## A.1 ALGORITHM

---

**Algorithm 1** Uniformization

---

**Require:** Point sets $P \in \mathbb{R}^{N \times 2}$, parameters $\alpha$, $\beta$, number of nearest neighbors $k$, boundary radius $R_b$, convergence threshold $\varepsilon$, maximum iterations $T_{\max}$

**Ensure:** Optimized point set $P'$

1: Initialize $P' \leftarrow P$
2: $t \leftarrow 1$
3: **repeat**
4:     Build KD-tree for point set $P'$
5:     **for** each point $i = 1$ to $N$ **do**
6:         Find $k$ nearest neighbors $\mathcal{N}_k(i)$ for point $\mathbf{p}_i$
7:         Compute repulsive force:
8:         $\mathbf{F}_i = \alpha \sum_{j \in \mathcal{N}_k(i)} \frac{\mathbf{p}_i - \mathbf{p}_j}{\|\mathbf{p}_i - \mathbf{p}_j\|(\beta + \|\mathbf{p}_i - \mathbf{p}_j\|^2)}$
9:     **end for**
10:     **for** each point $i = 1$ to $N$ **do**
11:         Update position: $\mathbf{p}_i^{(t+1)} = \mathbf{p}_i^{(t)} + \mathbf{F}_i$
12:         **if** $\|\mathbf{p}_i^{(t+1)}\| > R_b$ **then**
13:             Project to boundary: $\mathbf{p}_i^{(t+1)} \leftarrow R_b \cdot \frac{\mathbf{p}_i^{(t+1)}}{\|\mathbf{p}_i^{(t+1)}\|}$
14:         **end if**
15:     **end for**
16:     Compute total force: $F_{\text{total}} = \sum_{i=1}^{N} \|\mathbf{F}_i\|^2$
17:     $t \leftarrow t + 1$
18: **until** $F_{\text{total}} < \varepsilon$ or $t > T_{\max}$
19: **return** $P'$

---

In our experiments, we set $\alpha = 220$, $\beta = 150$, $k = 50$, $R_b = 900$, $\varepsilon = 0.1$, and $T_{max} = 200$.

## A.2 PROOF OF THEOREM 3.2

*Proof.* Since $\nabla U$ is $L$-Lipschitz on $K$, the standard descent lemma gives for any $X \in K$ and $0 < \eta \leq 1/L$,

$$U(X - \eta \nabla U(X)) \leq U(X) - \frac{\eta}{2}\|\nabla U(X)\|^2. \tag{1}$$

Applying this to $X = P^t$ yields

$$U(\tilde{P}^{t+1}) \leq U(P^t) - \frac{\eta}{2}\|\nabla U(P^t)\|^2.$$

Projection onto the closed convex set $K$ cannot increase the objective, hence

$$U(P^{t+1}) = U(\Pi_K(\tilde{P}^{t+1})) \leq U(\tilde{P}^{t+1}) \leq U(P^t) - \frac{\eta}{2}\|\nabla U(P^t)\|^2.$$

If $\nabla U(P^t) \neq 0$, then the right-hand side is strictly less than $U(P^t)$, implying that $U(P^{t+1}) < U(P^t)$. If $\nabla U(P^t) = 0$, the algorithm reaches a stationary point and the sequence remains constant. Since $U(P)$ is bounded below on $K$ (e.g., $U(P) \geq 0$ by construction), the sequence $\{U(P^t)\}$ is strictly decreasing until a stationary point is reached and thus convergent by the monotone convergence theorem. $\qquad\square$

