# OpenReview forum: "Latent Space Uniformization in Generation"
_ICLR.cc/2026/Conference — Submitted to ICLR 2026_

### Official Review · Reviewer_FYGM · 2025-10-27

**Soundness:** 2
**Presentation:** 2
**Contribution:** 2
**Rating:** 4
**Confidence:** 4

**Summary:**

This paper maps latent codes into 2D, applies a Coulomb-inspired repulsion to make the space more uniform, and then maps back the shifts to high dimensions for generation. The goal is to achieve smoother and more consistent cross-domain translations by regularizing the latent geometry.

**Strengths:**

1. **Creative and unconventional use of physical analogies**
   The idea of applying Coulomb’s law to model latent-space uniformization is genuinely creative. It’s rare to see such a direct, physically inspired formulation in modern generative modeling. This originality reflects open and out-of-the-box thinking, and the field benefits from more attempts to connect physical intuition with geometric latent design.

2. **Conceptually sound latent-space reasoning**
   The focus on operating in latent space, rather than image space, is a solid design decision. The canonical space representation and harmonic registration modules are conceptually coherent and help maintain semantic consistency across domains, which is supported by clear ablation evidence.

3. **Demonstrated improvement over immediate baselines**
   Results show moderate but consistent quantitative and qualitative improvements over prior methods such as GMapLatent and CycleGAN. FID and accuracy metrics, as well as ablations, demonstrate that each module (canonical representation and harmonic registration) contributes meaningfully to better semantic alignment and smoother translations.

4. **Coherent geometric interpolation mechanism**
   The curve-to-curve generation results effectively illustrate the intended advantage of smooth, continuous mappings through latent space. This geometric interpretation of interpolatio, particularly the spherical (Slerp-based) variant, successfully maintains semantic continuity and avoids abrupt mode changes during translation.

5. **Avoiding 2d bottleneck is elegant**
The fact that uniformization happens in the 2d space, but that the 2d representations are not the actual representations but only proxies for the uniformization considerations is elegant. Mapping the change to the higher dim rather than mapping the representation itself is a good useful solution.

**Weaknesses:**

1. **Fundamental non-scalability**
   The method by its own definition cannot scale.
   To be clear, this is not about the paper not showing higher-scale results. The algorithm itself depends on pairwise Coulomb-like repulsion and KD-tree construction in every iteration, with no batching or locality reduction.
   The complexity $O(T_{\max}(N \log N + kN))$ per cluster makes it fundamentally unsuitable for large datasets.

2. **Reliance on small, predefined class counts**
The paper never clarifies how clusters are obtained. From the description and examples, it appears that semantic labels are used directly, meaning the method is supervised rather than unsupervised. The method assumes a small number of semantic clusters that are known in advance. The angular sector layout and barycentric formulation both rely on these being fixed manually. This limits applicability to datasets with few discrete classes and cannot naturally extend to many or continuous domains. It seems there is a restriction allowing only annotated datasets.

3. **Angular-only uniformization bias**
   The “uniformization” process equalizes only the angular component, leaving the radial distribution unbalanced. Because the area element in polar coordinates scales as $r\,dr\,d\theta$, the resulting density grows roughly as $1/r$, producing a strong bias toward the center.
   This effect is visible even in the paper’s own illustrations, where samples appear clearly denser near the origin. I couldn't find any comment regarding this or even awareness. Basically this contradicts the premise of the paper because this is not uniform density, it is radial dependent density.

4. **Complexity omissions**
   While the paper provides complexity expressions for the repulsion updates, it omits the cost of clustering (when applicable) and likely undercounts repeated KD-tree rebuilds per cluster. The analysis therefore understates the actual computational cost.

5. **Presentation of Coulomb’s law**
   A very minor and friendly remark: “Theorem 3.1 (Coulomb’s Law)” is presented as if it were an original result.
   This is not how such known laws are usually included in ML papers—unless, of course, the author happens to be Charles-Augustin de Coulomb 🙂.

6. **Unclear definition of contributions**
   The “Contributions” paragraph does not clearly describe what was actually introduced or achieved relative to prior work.
   A contribution section should explicitly state what you did that was not done before. The current phrasing mixes broad intentions with unclear tasks, which makes it difficult for readers to identify the paper’s concrete novelty.
   I know this sounds picky, but here it was genuinely important for understanding what the authors consider their contribution.

7. **Weak and outdated results**
   The reported FID scores are modest (around 13.7 for digits and 81 for animals) and the claimed improvements over prior work are marginal well within noise.
   Comparisons are limited to older GAN-based methods (CycleGAN, MUNIT, StarGAN) and to Zeng et al. (2025).
   No modern diffusion-based or transformer-based translation models are included, despite these being the true state of the art.
   Therefore, the claim of “state-of-the-art” performance is outdated and not substantiated. The evaluation also relies on small, low-resolution datasets with no variance analysis, further limiting credibility.

**Questions:**

1. Could you do it in batches? of course the nearest-neighbor would become weird but this should be a key direction towards a feasible method.


I find the use of physics compelling, there is an elegant side of uniforming by density.
However, this paper seems to not be ready. To my understanding, currently the method is not feasible beyond toy data. Comparisons are only to old works and it feels like the writing of the paper is not great. Unfortunately, I don't think this paper should be accepted. I do encourage to keep perusing this direction and try to get it to something scalable.

**Details Of Ethics Concerns:**

I

---

### Official Review · Reviewer_ZRL3 · 2025-10-29

**Soundness:** 2
**Presentation:** 1
**Contribution:** 2
**Rating:** 2
**Confidence:** 3

**Summary:**

The paper proposes a new method to build an auto-encoder with a regularized and uniformly distributed latent space. This is done by a method that models latent points as charged particles driven by Coulomb forces, and that seeks equilibrium on this ensemble of particles. The authors apply experiments on single-domain and cross-domain generative models using their proposed method.

**Strengths:**

* Propose a novel method based on Coulomb potentials to make latent space uniformly distributed.

**Weaknesses:**

* The paper is not well written and lacks crucial details. It is vague about core components of the method. There is no algorithm of the overall method, no code provided, nor no details (e.g. hyper-parameters) on neural networks' trainings. For example, how is the encoder/decoder training combined with the latent space uniformization proposed by the authors. How does the cross-domain setting differs from the single-domain setting? Where do the categories come from in the barycentric translation: datasets' classes?

* There is also a lack of precise mathematical notation. There is no harmonized notations between sections. And many sections lack precise mathematical definitions of the operations that are performed, making the explanations of the methods very unclear. Overall, the quality of presentation is far from reaching the bar of ICLR's standards.

* Experimental results are not particularly strong. The method does not show strong advantages compared to concurrent methods.

**Questions:**

* Is the method limited to 2D latent spaces?

* What is the complexity and computational cost of latent space uniformization?

---

### Official Review · Reviewer_zKDr · 2025-10-29

**Soundness:** 1
**Presentation:** 1
**Contribution:** 1
**Rating:** 0
**Confidence:** 4

**Summary:**

This paper proposes a method to map a latent variable to a canonical manifold structure. This is motivated by the idea that the space from several different models can be aligned to support cross-domain generation and that having a simple structured latent representation will allow for simpler and more intuitive transformations. The approach taken in the paper is to first map the latent representation to two dimensions then transform this data to a clustered representation based on categorical information. This representation is then made uniform into a canonical representation.

Experimental results are shown in simple MNIST style examples comparing the proposed approached with a GAN approach.

**Strengths:**

The paper is somewhat original especially in the context of modern generative models. The task that the authors try to address is important.

**Weaknesses:**

This paper suffers from a rather chaotic disposition and it is hard to extract what exactly the authors are proposing and even after a few reading it a few times I find the paper confusing. The main issue I have with the paper is that I don't find any justification for what is proposed except for that it makes intuitive sense and that the results shows that it works. Sadly the experimental evaluation in a paper with these complex models will never be sufficient to draw conclusions from. To that end it is really hard to draw conclusions from this work.

Minor thing, but as you correctly say, diffusion models do not reduce the dimensionality of the observed data, still you refer to latent generative models as models that map from a low-dimensional latent representation.

**Questions:**

Can you clarify the use of UMAP?

---

### Official Review · Reviewer_oB2n · 2025-11-04

**Soundness:** 3
**Presentation:** 3
**Contribution:** 3
**Rating:** 4
**Confidence:** 3

**Summary:**

The paper proposes using uniformization and canonical representation to enable cross-domain generation. The idea is novel and interesting. The authors conduct experiments on both MNIST (mapping Chinese numerals to Arabic numerals) and on color images (transferring from one animal species to another). However, I have practical concerns and would like the authors to clarify the motivations as well as demonstrate the performance improvements achieved by their method. Please see below.

**Strengths:**

1. The core idea of the paper is novel and interesting, particularly the uniformization and alignment components.

2. The proposed algorithms do not require training an additional cross-domain generator, and the transfer process is geometric. This may improve computational efficiency.

3. The structure provides some explainability. Since the alignment is achieved through convex partitions and harmonic maps, it becomes possible to reason about where a specific sample will be mapped, which contrasts with the black box of large conditional diffusion models.

4. The paper is clearly written and easy to follow.

**Weaknesses:**

1. The algorithm heavily depends on a 2-D projection. However, in image generation, a 2-D representation is often insufficient to capture the full information of high-dimensional images. When projecting an original 256*256 space into only two dimensions, different categories of images may overlap significantly. In such cases, the algorithm may fail to properly distinguish classes during uniformization.

2. The improvement of ULatent over the existing method GMapLatent appears limited, as shown in Table 1. Although the authors mention that GMapLatent has higher computational complexity, the paper does not provide comparisons of training or inference time to support this claim.

3. Although the method does not require training a separate cross-domain generator in the latent space, it still requires training separate VAEs for each image class (based on my understanding). If this is true, then when the number of classes becomes large, the computational cost may increase significantly and offset the claimed efficiency benefits.

**Questions:**

Following up the weaknesses section, I have some questions that I want the authors to clarify:

1. Could the authors justify whether the proposed algorithm can still succeed when the number of classes is large and when the image dimensionality increases? Please explain why, and under what conditions, the approach would remain effective.

2. How does this method compare to conditional diffusion models operating in the latent space of a VAE, such as Stable Diffusion or the more recent conditional diffusion/flow approaches trained over VAE latent spaces ([1]–[3])? These methods only require training a single VAE across all classes and an additional conditional diffusion model in the latent space.

3. Could the authors provide training and sampling efficiency comparisons against other baselines, especially GMapLatent?


[1] Xu, Chen, Xiuyuan Cheng, and Yao Xie. "Computing high-dimensional optimal transport by flow neural networks." arXiv preprint arXiv:2305.11857 (2023).

[2] Batzolis, Georgios, et al. "Conditional image generation with score-based diffusion models." arXiv preprint arXiv:2111.13606 (2021).

[3] Melistas, Thomas, et al. "Benchmarking counterfactual image generation." Advances in Neural Information Processing Systems 37 (2024): 133207-133230.

---

### Official Review · Reviewer_MVQz · 2025-11-12

**Soundness:** 3
**Presentation:** 3
**Contribution:** 2
**Rating:** 4
**Confidence:** 3

**Summary:**

This paper's core idea, ULatent (Latent Space Uniformization), uses Coulomb's Law to get the messy latent space organized. They treat all the data points as identical charged particles that repel each other until they are spread out uniformly in a canonical structure, eliminating useless empty areas. The benefit is huge: sampling is better (every random spot gives a meaningful output), and cross-domain translation (like changing a horse to a zebra) is more accurate because they use a Barycentric Translation trick to precisely align features. The big question, though, is that they only proved this dynamic, particle-repulsion method works in a 2D space, making it unclear if it can handle the much higher dimensions used by modern, serious AI models.

**Strengths:**

1. Using Coulomb's Law to fix a deep learning problem is interesting. It makes the whole complex math easy to grasp: they just make the points hate each other until they're perfectly spaced.

2. The paper is aiming to fix the two biggest annoyances in generative models: making sure you can sample anywhere and still get a decent picture (no more dead zones), and making cross-domain translations (like cat to dog) reliable.

3. The writing is good. E.g. The main algorithm (A.1) is laid out step-by-step. It's easy to look at and figure out how to rebuild it yourself.

**Weaknesses:**

1. The paper only modeled this whole repulsion thing in a 2D latent space. This could make the entire method feel like a toy problem or just a cool concept that might completely fail due to the Curse of Dimensionality when scaled up to real-world complexity.

2. Their goal is purely geometric which make the points evenly spread out. But forcing points away from their neighbors doesn't mean they'll land in a semantically useful area. They might get pushed into a noise zone or an area that generates a bad image, just so they can maintain distance. The method might actually break the subtle semantic structure the model learned.

**Questions:**

1. The work only showed results in a 2D latent space. What happens when we jump to practical dimensions (128D/512D)? Will the method even scale, or will the k-NN cost and the Curse of Dimensionality totally break the uniformization process?

2. Why the weird, custom force formula? It’s not standard Coulomb's Law, and it relies on arbitrary-looking $\alpha$ and $\beta$ parameters. Can you provide a strong theoretical reason for this specific modification and prove its stability?

3. The proposed method is purely geometric repulsion. How do you ensure this uniform spreading doesn't push points into dead zones or break the semantic structure learned by the original encoder/decoder? Does uniform geometry actually hurt content integrity?

4. Is this slow, iterative process a one-time post-processing step? If so, how do you handle the encoder? If it's a regularizer during training, how much does it slow down model convergence? The cost seems prohibitive either way.

---

### Author Response · Authors · 2025-12-04
**Clarifications on 2D Latent Space Uniformization for Cross-Domain Alignment**

We deeply appreciate your insightful comments and suggestions. The responses to the common questions from reviewers are given as follows. We clarify that the proposed uniformization process works on 2D latent spaces; its main purpose is to precisely and seamlessly align 2D latent spaces, and therefore build the correspondence between high-dimensional latent spaces. The whole framework is designed for solving cross-domain multi-class translation problems.

**The Use of UMAP (To zKDr):** UMAP is employed to achieve bidirectional mapping between high-dimensional and 2D latent codes. Our approach operates on 2D UMAP latent codes and build the correspondence between latent spaces. The specific workflow is as follows: high-dim code→UMAP→ULatent→UMAP→high-dim code. This enables cross-domain alignment without altering the distribution in the high-dimensional space.

**Scale & Dimensionality (To MVQz, oB2n and ZRL3):** Our method establishes coordinate indices for high-dimensional data through 2D latent space. By sacrificing controlled semantic distortion, it achieves breakthrough efficiency in neighborhood computation and cross-domain transformation. Utilizing 2D grids or KD-trees enables rapid nearest neighbor retrieval, thereby circumventing the curse of dimensionality inherent in high-dimensional spaces. This approach alters visual effects and generates novel outcomes without compromising the original content structure, and simultaneously preventing mode collapse due to the seamless distribution.

The computation of uniformization requires 2D latent codes (by UMAP and encoder) as input, therefore the image dimensionality won’t affect the algorithm efficacy. The number of classes neither get involved in the computation; only the number of points and their initial distribution affect the computation efficiency.

**Coulomb's Law (To MVQz):** Our formula is derived from Coulomb's law and shall be renamed as a lemma within this paper in the final version. Its core advantage lies in delivering exceptional numerical stability. By employing the arctangent function, it eliminates the singularity present in the standard law as $ r \to 0 $. This prevents gradient explosions and renders the formula insensitive to noise and outliers. A detailed proof of convergence in Appendix A.2 provides a theoretical analysis for its stability.

**Complexity & Cost (To MVQz, oB2n and ZRL3):** Our method is a one-time, plug-and-play postprocessing technique. It operates within a 2D latent space, without altering the original training process of the encoder and decoder. We think incorporating uniformization as a regularization term into training is inappropriate. As the training aims to learn the original data structure, the objective of uniformization is to achieve efficient, mode-collapse-free alignment and generation. Performing uniformization calculations directly in high-dimensional space is computationally expensive and suffers from convergence issues, whereas in 2D space the computational cost is manageable and the approach is feasible.

Regarding uniformization computation, compared to the $O(N^{3})$ time and $O(N^{2})$ space complexity of optimal transport methods, our uniformization approach employs approximate nearest neighbor search to maintain complexity at $O(kNlogN))$ for time and $O(Nk)$ for space (where $k$ denotes the specified number of nearest neighbors). The computation time for uniformization via KD-tree batch processing is 17s for Chinese-MNIST (8k pts) and 103s for MNIST (59k pts) on AMD 4600H with 3GHz Raden Graphics.

**Comparison (To oB2n):** Our framework focuses on cross-domain conversion without mode collapse. The proposed module can be integrated into the existing methods which involves latent space as a plug-in to the 2D latent space (generated by UMAP of high-dimensional latent space). This does not directly compete with the conditional diffusion model, but rather provides it with a latent space foundation featuring a more optimal geometric structure; the two can work in tandem.

Unlike optimal transport resampling and nearest neighbor assignment in the GMapLatent model, our approach efficiently computes uniformization while preserving the original points in the 2D latent space. This enables more precise interpolation by respecting genuine neighborhood information.

**Feasibility & Practicality (To FYGM):** To make large-scale uniformization practical, we use batch processing. We let a KD-tree find neighbors for all points at once, then compute everything in parallel. This approach does loosen the coordination between points when searching for neighbors, but it's a worthwhile trade-off. It makes about 10 times faster and truly feasible for big datasets.

Thank the reviewers and organizers for your valuable comments and insights, which helped us more comprehensively view our work. We deeply appreciate it if you find our responses worthy of a higher score. Thank you for your support!

---

### Meta-Review · Area_Chair_6hRV · 2025-12-12

**Summary:**

Unanimity of the reviewers on the negative side. The authors provided responses to a number of concerns. The paper requires major revision and a second review round. Under its current form, the paper can not be accepted for publication.

**Reviewer Concerns:**

Unanimity of the reviewers on the negative side. The authors provided responses to a number of concerns. The paper requires major revision and a second review round. Under its current form, the paper can not be accepted for publication.

**Reviewer Scores:**

Unanimity of the reviewers on the negative side. The authors provided responses to a number of concerns. The paper requires major revision and a second review round. Under its current form, the paper can not be accepted for publication.

---

### Decision · Program_Chairs · 2026-01-26

Reject